# Cytokine Profile of Invasive Pulmonary Aspergillosis in Severe COVID-19 and Possible Therapeutic Targets

**DOI:** 10.3390/diagnostics12061364

**Published:** 2022-06-01

**Authors:** Alessandro Russo, Helen Linda Morrone, Salvatore Rotundo, Enrico Maria Trecarichi, Carlo Torti

**Affiliations:** Infectious and Tropical Disease Unit, Department of Medical and Surgical Sciences, “Magna Graecia” University of Catanzaro, Viale Europa, 88100 Catanzaro, Italy; helen.morrone@gmail.com (H.L.M.); srotundo91@gmail.com (S.R.); em.trecarichi@unicz.it (E.M.T.); torti@unicz.it (C.T.)

**Keywords:** COVID-19, IPA, CAPA, cytokines, therapy

## Abstract

During the SARS-CoV-2 pandemic, a higher incidence of invasive pulmonary aspergillosis was observed in patients affected by Coronavirus disease 2019 (COVID-19), leading to the delineation of a new entity named COVID-19 associated pulmonary aspergillosis (CAPA). A predisposition to invasive infection caused by *Aspergillus* spp. in SARS-CoV-2 infected patients can be ascribed either to the direct viral-mediated damage of the respiratory epithelium, as already observed in influenza H1N1 virus infections, or to the dysregulated immunity associated with COVID-19. This narrative review focuses on the impact of immune impairment, particularly due to cytokine dysregulation caused by *Aspergillus* spp. superinfection in COVID-19 for a more in-depth understanding of the molecular pathways implicated in CAPA. As immune competence has proven to be essential in protecting against CAPA onset, a role already threatened by SARS-CoV-2 infection itself, preventive strategies should focus on reducing factors that could further target the host immune system. We also aimed to focus on well-known and less-known risk factors for IPA in COVID-19 patients, related to the main causes of immune suppression, both virus-mediated and iatrogenic, including treatments currently indicated for COVID-19. Lastly, possible preventive strategies aimed at reducing morbidity and mortality due to CAPA could be implemented.

## 1. Introduction

SARS-CoV-2 is an airborne coronavirus which can infect a variety of cells, such as lung alveolar, endothelial, immune, cardiac, and renal cells, by binding angiotensin-converting enzyme 2 (ACE-2) [1]. Following entry into the cells, the virus can determine an important release of inflammatory cytokines, which can lead to the dysfunction of the vascular epithelium, thrombosis, and an immune-deficient status, thus facilitating bacterial and/or fungal superinfections [2,3].

In critically ill Coronavirus disease 2019 (COVID-19) patients, many additional risk factors can affect the competence of the host immune system, such as invasive ventilation, sedation, the impairment of mucociliary clearance, and the administration of immunosuppressive drugs [4]. Furthermore, approximately 5–30% of COVID-19 patients require intensive care [5] and, interestingly, in this setting, invasive pulmonary aspergillosis (IPA) can reach a prevalence of up to 4–30% [6].

During the influenza H1N1 pandemic in 2009, a higher incidence of IPA was observed in infected patients, particularly those admitted to the ICU; direct virus-mediated damage to the airway epithelium as well as cell-mediated damage and disruption of mucociliary clearance were suggested as the main mechanisms responsible [7]. A similar phenomenon was evidenced during the COVID-19 pandemic in patients infected by SARS-CoV-2, leading to the delineation of a new entity named COVID-19 associated pulmonary aspergillosis (CAPA) [7].

In this narrative review, we aimed to investigate the possible molecular mechanisms underlying CAPA by examining both cytokine profiles that characterize COVID-19 pneumonia and CAPA; we also aimed to investigate possible causes associated with CAPA in patients with COVID-19 in terms of predisposing factors and hypothesize effective strategies to prevent CAPA, as well as possible treatments for this life-threatening condition.

## 2. Invasive Pulmonary Aspergillosis in COVID-19: Where We Are Now and Where We Are Going to

Lethality from CAPA ranges from 50% (in COVID-19 affected subjects with proven or probable IPA) to 80–90% (in patients in which invasion of blood vessels by the mold occurs), as evidenced by positive serum and/or respiratory galactomannan. Therefore, CAPA is a life-threatening disease [8,9].

It is difficult to diagnose and manage CAPA. First, *Aspergillus* spp. is ubiquitous in the environment, and often conidia inhaled by the immunocompetent host do not germinate and are rapidly cleared from the airways without causing infection [9]. For this reason, the definitive diagnosis of IPA is difficult to obtain, so it is frequently expressed in terms of likelihood as probable or possible rather than proven [9]. Importantly, only a few patients display classical risk factors such as the use of corticosteroids or chronic pulmonary disease [9], suggesting that acute distress respiratory syndrome (ARDS) due to severe COVID-19 by itself or the associated immune impairment may be independent risk factors [10].

The prevalence of CAPA is widely variable, ranging from 5%, in the medical area, to 30%, in intensive care settings [11]. This variability can stem from many causes: environmental factors (which are variable by seasons and geographical areas); diagnostic tools with different characteristics in terms of sensitivity and specificity (with galactomannan used both as a screening test or additional diagnostic tool); timing and performance of bronchoscopy; reduced sensitivity in samples such as bronchial aspirate or sputum compared with bronchoalveolar lavage (BAL); lastly, the use of molecular tests to identify *Aspergillus* spp. has to be implemented [12]. Not surprisingly, CAPA appears to be much more frequently diagnosed at autopsy [13]. Overall, these observations emphasize a possible selection bias, which makes it difficult to extrapolate real prevalence data [14].

Many host-related or environmental risk factors are implicated in the pathogenesis of invasive aspergillosis. Neutropenia, hematologic malignancies, allogenic bone marrow transplantation, solid organ transplantation, neoplasm, and HIV infection are classical host-related risk factors for IPA, while climatic variables such as airborne mold concentration, geographic area, remodeling, or construction work are considered important environmental factors [15]. More recently, new risk factors were identified, such as end-stage COPD requiring chronic high-dose steroid therapy, Child–Pugh C liver cirrhosis, and immunosuppressive therapies (i.e., monoclonal agents, checkpoint inhibitors, steroid therapy) [10].

Despite the important efforts to characterize the molecular pathways involved in SARS-CoV-2 and *Aspergillus* spp. co-infections, more studies are needed on immune phenomena associated with this life-threatening condition. Understanding the risk factors of CAPA could lead clinicians to target screening of some patients or even use antifungal prophylaxis in selected groups of patients.

## 3. Cytokine Storm in Severe COVID-19

Severe COVID-19 is characterized by an uncontrolled and excessive release of cytokines which causes the hyperactivation of the immune system (known as a “cytokine storm”) [16]. Paradoxically, this extreme activation of the immune system results in a state of immune paralysis and immune deficiency, in some cases associated with a severe clinical condition known as hemophagocytic syndrome [17]. Signs and symptoms characterizing the hemophagocytic syndrome include fever, thrombocytopenia, low red and white blood cell count, and hepatosplenomegaly, while laboratory findings are hyperferritinemia, hypertriglyceridemia, and low fibrinogen [18].

SARS-CoV-2 severe infection is characterized by the high expression and release of cytokines (such as TNF, IL-1, IL-6, IL-8) and a low expression of interferon-γ [19]. In COVID-19 pneumonia, high TNF, IL-6, and IL-8 levels are associated with poor outcomes [20]. In patients with COVID-19 requiring intensive care, low CD4+ and CD8+ cell counts are also detected, negatively correlated with a high expression of cytokines (IL-6, IL-10, and TNF-α). Furthermore, an inverse correlation of low levels of serum IL-6, IL-10, and TNF-α with the normalization of T cells count was seen in patients with clinical resolution of the disease [21]. These data, supported by studies on animal models, suggest that a cytokine storm suppresses adaptive immunity against SARS-CoV-2 infection, also determining functional exhaustion in those lymphocytes that remain [21,22]. We hereby analyze the most relevant cytokines involved in SARS-CoV-2 infection.

IL-1α is a constitutively expressed cytokine released by necrotic cells and acts as an alarmin, signaling the loss of membrane integrity and thus orchestrating the development of an inflammatory response [23]. IL-1α plays a central role in the pathogenesis of ARDS in COVID-19: the release of IL-1α after the damage of the lung epithelium results in myeloid cell recruitment and inflammasome activation, leading to amplification of the inflammatory cascade. This cytokine is also involved in the promotion of inflammatory thrombosis [19]. However, these mechanisms are counter-regulated by the IL-1 receptor antagonist (IL-1Ra) which is released to limit the excessive inflammatory process; in fact, it has been found in high concentrations in the BAL of patients with ARDS, especially in the resolution phase [24].

IL-6 is a cytokine with pleiotropic functions, released by myeloid cells and involved in the acute immune response to infection, as well as in the pathogenesis of autoimmune disease [25]. IL-6 is significantly elevated in severe COVID-19 patients [26] and is associated with an increased risk of thrombosis, organ damage, and death [2]. Patients with IL-6 levels above 50–80 pg/mL are at higher risk for mortality and severe disease [27]. In patients who died of COVID-19, IL-6 levels were higher than in surviving patients [28].

IL-8 appears to be the cytokine that characterizes severe SARS-CoV-2 pneumonia; in fact, it does not increase in mild/moderate COVID-19 [29], while in severe COVID-19 pneumonia, high levels are associated with a worse degree of respiratory failure and higher mortality [20]. Alveolar neutrophils derived from patients with severe SARS-CoV-2 pneumonia show increased transcription of IL-8 mRNA and, if incubated with IL-8, further production is observed, which causes the attraction and activation of neutrophils in loop-like feedback. Moreover, this neutrophil-IL-8 dysregulation has been associated with a prothrombotic-neutrophilic phenotype, characteristic of severe COVID-19 [29].

IL-10 is a cytokine with anti-inflammatory and immunosuppressive effects, exerted via the inhibition of antigen presentation by macrophages and dendritic cells, the limitation of T lymphocyte proliferation, and suppression of proinflammatory cytokine synthesis [30]. Interestingly, an early increase in IL-10 is observed in patients with severe COVID-19; this could be explained both by the fact that IL-10 may not be able to properly modulate the immune response in severe SARS-CoV-2 infection, as seen in diabetic subjects who are known to have more severe disease [31] or IL-10 could deviate from its classical immunomodulatory action and act as an immunostimulant cytokine in some diseases as well as in severe COVID-19 [32].

## 4. Cytokine Expression in IPA

In immunocompetent hosts, the key defense against *Aspergillus* spp. infection is represented by the recruitment and activation of cellular immunity, specifically neutrophils and monocytes capable of engulfing and killing the conidia [33]. Neutrophils also neutralize the fungal hyphae through degranulation and release of oxidants, and a lack of adequate innate immune response, as occurs in neutropenic patients, may lead to conidial germination and, eventually, invasive infection [34].

TNF plays a pivotal role in protecting the lungs from *Aspergillus* spp. as demonstrated by the increased risk of IPA in patients treated with anti-TNF drugs such as infliximab [35]. In fact, TNF is markedly expressed in the lungs following inhalation of *Aspergillus* spp. conidia; moreover, TNF neutralization results in impaired fungal clearance and increased risk of death [36]. Exogenous TNF protects immunodeficient mice from IPA [37].

The administration of INF-γ in immune-deficient mice has proven to be protective against IPA. The underlying mechanism may be explained by the ability of INF-γ to enhance NK lymphocyte fungicidal activity against A. fumigatus [37].

As reported above, IL-6 is a crucial cytokine in the activation of innate immunity and has been associated with protection against *Aspergillus* infection [20]. Proteases secreted by A. fumigatus can induce the transcription of IL-6 and IL-8 in the lung epithelial cells via the activation of the NFkB pathway, thus stimulating the immune response [38].

Cytokine serum levels in *Aspergillus* spp. infection may also have a predictive value in terms of the progression of disease and therapy response: patients who are clinical responders to antifungal therapy show a decrease in IL-6 serum levels, whereas non-responders maintain an elevated concentration of the cytokine. IL-10 is an anti-inflammatory cytokine secreted by Th2 and T-regulatory lymphocytes, whose function is to dampen the inflammatory cascades [34]. It has been shown that IL-10 deficiency protects against pulmonary aspergillosis [39]. One reason may be that macrophages exposed to high concentrations of IL-10 lose the ability to produce superoxide and thus are incapable of clearing the hyphae of *Aspergillus* spp. [40]. In transplanted patients with stem cells, high levels of IL-10 hindered the elimination of the fungus without inhibiting the ability to engulf *Aspergillus* spp. [41]. High IL-10 baseline levels are associated with survival; conversely, a high baseline level of IL-8 correlates to poor clinical response and death [34]. Patients without IPA, compared with patients with probable IPA, show significantly lower levels of IL-6 and IL-8 in both serum and BAL, and these two cytokines positively correlate with galactomannan levels [42]. Finally, IL-17 expression by neutrophils is increased during *Aspergillus* spp. infection [43].

Many studies on murine models showed the important role of IL-18 and IL-12 in early immune response and fungal clearing in *Aspergillus* spp. infection. IL-12 knockout and IFN-γ knockout mice had an enhanced susceptibility to IPA, while IL-4 knockout models showed a resistance to the infection, presumably due to a Th-1 cytokine phenotype, confirming the protective effect of these cytokines in IPA. Conversely, IL-10 deficient mice showed an enhanced Th-1 mediated response and less severe disease [44].

Although the underlying molecular pathways in CAPA were not completely disclosed, it is possible to hypothesize the main role played by COVID-19 is associated with the cytokine storm, with an elevation of IL-1, IL-6, IL-8, and IL-10, that leads to immune-dysregulation and lymphocyte exhaustion, thus facilitating the onset and progression of IPA (see Figure 1).

## 5. Genetic Susceptibility to *Aspergillus* Infection: Possible Role in CAPA

As mentioned above, innate immunity plays an essential role in battling invasive aspergillosis. Genetic factors may contribute to the altered immune host defense that predisposes to the fungal infection. For example, a genetic deficiency in NADPH-oxidase gives a strong predisposition to IPA [45]. Single nucleotide polymorphisms in Toll-like receptors are associated with higher susceptibility to noninvasive pulmonary aspergillosis, particularly lower macrophage expression of TLR-3 and TLR-10 [46,47]. Moreover, specific polymorphisms in genes coding for VEGF, IL-15, TREM-1, and PLAT are prevalent in patients with chronic cavitary pulmonary aspergillosis [47]. Different allele expressions and genotypes of mannose-binding lectin and surfactant proteins (SPA1 and SPA2) are found in patients with cavitary pulmonary aspergillosis and allergic bronchopulmonary aspergillosis, suggesting that genetic factors may predispose the patient to different manifestations of the disease [48].

Regarding COVID-19, studies have shown that genetic and epigenetic factors involving ACE-2, Toll-like receptors, HLA, interferon, and interleukin coding genes are associated with different outcomes of disease [49]. However, few data are reported about genetic predisposition in CAPA [50], and future studies should eventually confirm the role of polymorphisms in those genes and its implication in the innate immune response that may predispose to the development of CAPA in susceptible patients.

## 6. Impact of COVID-19 Therapy on CAPA

Steroids are the backbone of SARS-CoV-2-related respiratory failure treatment; the National Institute of Health (NIH) recommends a regimen of dexamethasone 6 mg/day for up to 10 days in COVID-19 hospitalized patients who require supplemental oxygen [51]. However, as mentioned above, glucocorticoid treatment is a known risk factor for *Aspergillus* spp. infection in non-neutropenic subjects. Steroids have an anti-inflammatory action mainly mediated by the suppression of the transcription factor NF-kB and are capable of increasing lung concentrations of IL-10, thus reducing the natural immune defenses against molds, i.e., macrophage activation and conidial phagocytosis and clearance [52]. Conversely, they do not inhibit neutrophil recruitment at the site of infection: activated polymorphonuclear cells attracted by IL-8 signaling are responsible for lung tissue damage in steroid-treated patients with *Aspergillus* spp. infection [53]. This is a different mechanism compared to neutropenic patients with IPA: in the latter case, there is no immune-mediated damage to the lung parenchyma, and the hyphae invade the blood vessels directly [53].

Additionally, in vitro and in vivo studies have shown that glucocorticoids act as growth factors for *Aspergillus* spp., as well as for other molds [54,55]. A daily dosage greater than 7.5 mg [56] and cumulative dosage of 100 mg [57] has been associated with an increased risk of developing CAPA.

Regarding antibiotics, azithromycin was initially used in severe COVID-19 for in vitro evidence of its antiviral effect, its anti-inflammatory properties, and to prevent bacterial superinfections [58]. However, azithromycin is associated with CAPA at a cumulative dose greater than 1500 mg. In fact, although azithromycin is able to reduce serum levels of IL-6, it inhibits the activity of neutrophils by reducing oxidative burst and favoring neutrophil apoptosis, both mechanisms inducing susceptibility to mold infection [57].

As a matter of fact, anti-TNF agents have an important anti-inflammatory effect and the ability to reduce the expression of some cytokines associated with poor outcomes in COVID-19 pneumonia [23]. As far as we know, there are no RCTs on the use of anti-TNF in severe COVID-19, and our knowledge derives from indirect observations of patients who are on anti-TNF drugs for underlying comorbidities and who contract SARS-CoV-2 infection; observational studies suggest that those patients tend to have a better outcome [59,60]. Therapy with anti-TNF antibody infliximab, however, is considered a major risk factor for IPA infection, with an even higher incidence described when the patient is in concomitant chronic steroid treatment [61].

The monoclonal antibody directed against INF-γ emapalumab, in combination with dexamethasone, was shown to be effective in remission of the primary hemophagocytic lymphohistiocytosis syndrome in pediatric patients [62] and its use has been suggested as a rescue therapy in patients with a refractory COVID-19-related cytokine storm [63]. There are no specific data in the literature about the correlation between the use of emapalumab and *Aspergillus* spp. infection.

Although direct blockade of TNF or INF-γ has not shown clinically useful results in improving COVID-19 prognosis so far, the downstream pathways activated by those two cytokines, such as the JAK/STAT1 pathway that play a central role in hyperinflammation and cell death, may be more promising targets [16]. Baricitinib is an inhibitor of the JAK/STAT pathway approved for the treatment of rheumatoid arthritis with good efficacy and safety profile [64] and has also been approved for use in COVID-19 hospitalized patients with rapidly increased oxygen need and secondary inflammation [65]. Ruxolitinib is a JAK 1/2 inhibitor indicated for myelodysplastic syndromes such as myelofibrosis and polycythemia vera [66]. It was successfully used in secondary hemophagocytic lymphohistiocytosis syndrome with a resolution of symptoms and laboratory abnormalities within the first two weeks of treatment [67] and subsequently to calm the cytokine storm due to severe COVID-19 [68]. Unfortunately, it has been shown that ruxolitinib inhibits IL-17 production and reduces hyphal killing of *Aspergillus* spp. by neutrophils [69].

Targeting IL-1 signaling could reduce hypoxia related to *Aspergillus* spp. infection and lead to improved outcomes of IPA [33,70]. Although anakinra, a monoclonal antibody directed against IL-1 receptor, has shown effectiveness in treating some forms of secondary haemophagocytic syndrome in pediatric patients [71], both anakinra and canakinumab, a monoclonal antibody directed against the ß subunit of IL-1, resulted in being safe but ineffective for the treatment of COVID-19 in a meta-analysis carried out on 16 RCTs [72]. It is still unclear whether anti-IL-1 can be beneficial in severe COVID-19 pneumonia, and there is no sufficient evidence to recommend for or against their use [51].

Tocilizumab, a monoclonal antibody directed against IL-6 receptor, is effective in suppressing the cytokine storm that leads to severe SARS-CoV-2 pneumonia [73]. It has been hypothesized that the efficacy of tocilizumab could also depend on the ability of the drug to intervene in coagulation with an antithrombotic effect in addition to its anti-inflammatory role [74]. Tocilizumab was associated with an increased risk of CAPA in ICU patients in a large multicenter study [75]. Although the indications for the use of tocilizumab and other immunosuppressive drugs in severe SARS-CoV-2 pneumonia have changed rapidly over time [51], it is difficult to clearly define whether or not these therapies result in an increased risk of CAPA and discontinuation of tocilizumab during CAPA is not currently recommended [11].

## 7. Preventive and Therapeutic Strategies for CAPA

When CAPA is suspected, antifungal treatment should promptly be started. Voriconazole remains the first-line treatment in IPA; however, it is associated with a high drug–drug interaction rate, hepatotoxicity, and QT interval elongation; the alternative use of isavuconazole may bypass such limitations, especially in intensive care settings [76].

An observational study evidenced that antifungal prophylaxis with posaconazole administered to COVID-19 patients admitted to ICUs may reduce the incidence of CAPA, although it does not show an impact on survival [77]. The role of antifungal prophylaxis in COVID-19 to prevent IPA is still to be determined. It could be hypothesized that severe COVID-19 patients with independent risk factors for IPA other than SARS-CoV-2 infection (i.e., hematologic malignancies, solid organ transplant, long ICU stay, immune-suppressive therapy) would likely benefit from prophylaxis more than other patients; indeed, it would be interesting to compare the impact of prophylaxis in COVID-19 patients stratified by a high or low risk of IPA. In this regard, serum levels of interleukins such as IL-6 and IL-8, associated with a high risk of IPA, could play a role as possible predictive biomarkers for higher risk of CAPA and thus be included in therapeutic decision-making [78,79]. The prevention of IPA should be a priority in the management of moderate-to-severe COVID-19 patients, particularly in ICU settings (see Figure 2). Prevention strategies include reducing risk factors for IPA to the minimum when feasible, for example, the length of invasive ventilation and ICU stay.

Regarding steroids, it is not currently recommended to discontinue steroid treatment in CAPA [11]. We consider, however, that risk and benefits should be taken into consideration when administering steroids in COVID-19 patients, additional risk factors for IPA should be weighed, and steroid therapy should not be, in any case, prolonged beyond necessary. A reasonable approach in patients with a high risk of developing CAPA could be to administer the lower dosage of steroid therapy recommended (i.e., desamethasone 6 mg/daily) and de-escalate as soon as possible.

As discussed above, immune-suppressant drugs that act by interfering with cytokine storm cascades, such as tocilizumab, baricitinib, and ruxolitinib, although potentially beneficial in reducing severe COVID-19 mortality and morbidity, are associated with a higher risk of IPA [69,75]. Therefore, their use should be withheld in patients with CAPA and those at high risk for CAPA.

Although anti-IL-1 agents such as anakinra and canakinumab are not recommended in COVID-19 pneumonia according to current guidelines [51], their effectiveness in reducing cytokine storms may make them suitable candidates as alternatives to tocilizumab and baricitinib in patients with severe COVID-19 and high risk of CAPA [33,70]. Further studies on the use of anti-IL-1 agents are required to provide evidence in favor of this hypothesis.

## 8. Conclusions

Similarly to the H1N1 influenza pandemic, the high incidence of *Aspergillus* superinfection in COVID-19 affected patients poses a major threat in terms of increased morbidity and mortality. The reasons underlying a predisposition to CAPA in COVID-19 patients are, as discussed, multiple and can be summarized as follows: individual patient susceptibility; COVID-19 pneumonia-related factors senso strictu; factors involving the in-hospital management of COVID-19 patients (i.e., prolonged ICU stay, invasive/noninvasive ventilation); use of immune-suppressive regimens recommended for the treatment of COVID-19 pneumonia. Patient susceptibility includes underlying comorbidities that predispose to IPA, chronic immune-suppressive treatment (i.e., immunomodulants or chemotherapy), and genetic factors that impair the immune system [80].

While SARS-CoV-2 infection facilitates *Aspergillus* spp. superinfection, the main predisposing factor for IPA involves immune system dysregulation that ensues in moderate to severe COVID-19 pneumonia. Firstly, the cytokine storm that occurs in COVID-19, instead of activating a competent immune response to possible opportunistic infections, causes the dysregulation of the immune system, with a reduction of lymphocyte count, dampening of cell-mediated fungicidal activity and ineffective conidial killing, creating a fertile ground for fungal invasion. Secondly, the cytokine pattern expressed in severe COVID-19 shares some similarities with severe IPA (i.e., high levels of TNF-α, IL-1, IL-6, IL-8, IL-10, and low levels of IFN-γ), as reported in Figure 1 [81]. Furthermore, steroids that are used as a backbone therapy for COVID-19 pneumonia and interleukin-targeting agents used to control cytokine storm (i.e., infliximab, tocilizumab, baricitinib, ruxolitinib) are known to be associated with a higher risk of *Aspergillus* superinfection, acting as immune-suppressants in an already dysregulated host immune response.

Considering the many factors that contribute to CAPA predisposition, preventive protocols toward fungal superinfection should be thought out and put into action in COVID-19 affected patients, especially those who are at higher risk of developing IPA [82]. The assessment of CAPA risk may include serum interleukin levels such as IL-6 and IL-8, which play a role in stratifying patients at higher and lower risk of fungal superinfection.

It is important to underline that diagnosis of CAPA is frequently presumptive, while proven diagnosis is infrequent in ICU patients. Consequently, the performance of the different non–culture-based tests have been evaluated, especially in ICU patients. As a matter of fact, the diagnostic performance of BAL galactomannan is superior to that of serum galactomannan, and the use of either BAL or serum 1,3-β-d-glucan shows suboptimal specificity. The performance of other non-culture-based tests (such as BAL *Aspergillus* lateral flow device and BAL/ blood *Aspergillus* PCR) is promising, but data are also limited in critically ill patients [83].

Of importance, management of steroid therapy is a milestone of CAPA treatment, suggesting that a lower dosage of steroids (dexamethasone 6 mg/daily) and an early de-escalation should be considered in patients with CAPA. Targeted therapies against interleukins approved for severe COVID-19 patients with acute worsening of respiratory failure (i.e., tocilizumab and baricitinib) should be avoided in patients with other independent risk factors for CAPA. Moreover, in patients with COVID-19 pneumonia treated with immune-suppressants or chemotherapy, risks and benefits should be carefully weighed during SARS-CoV-2 infection. Finally, although not yet approved for the treatment of patients with severe COVID-19 in alternative to anti-IL-6 agents, future research could assess the potential role of monoclonal antibodies targeting IL-1 (anakinra and canakinumab) in patients with cytokine storm and high risk of CAPA.

## Figures and Tables

**Figure 1 diagnostics-12-01364-f001:**
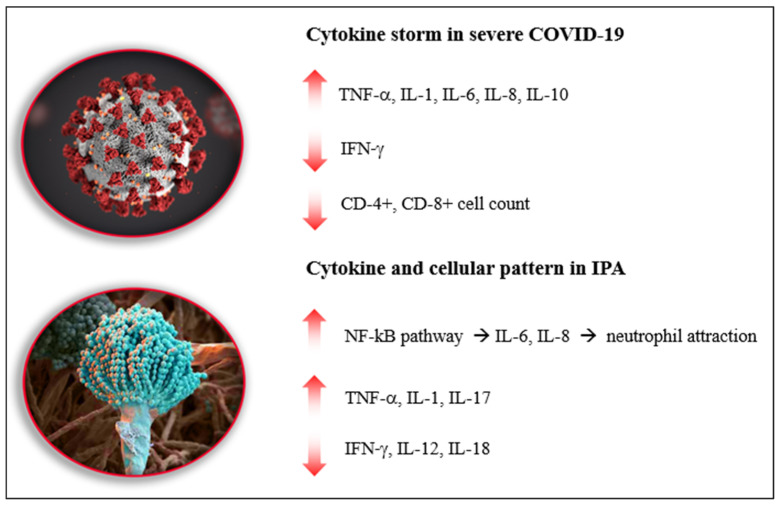
Cytokine expression during COVID-19 related cytokine storm and IPA–related immune dysregulation. Legend. COVID-19: Coronavirus disease 2019; CAPA: COVID-19 associated pulmonary aspergillosis; IPA: invasive pulmonary aspergillosis.

**Figure 2 diagnostics-12-01364-f002:**
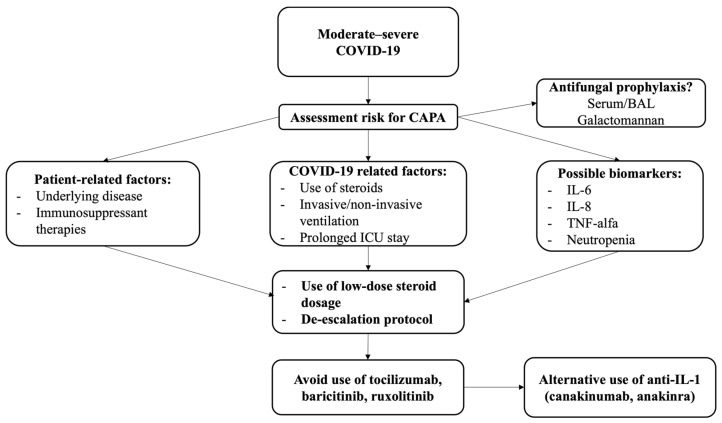
Strategies for the prevention of IPA in COVID-19. Legend. IPA: invasive pulmonary aspergillosis; CAPA: COVID-19 associated pulmonary aspergillosis; BAL: bronchoalveolar lavage; ICU intensive care unit; IL: interleukin.

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
