# Peer review of "Cytokine Profile of Invasive Pulmonary Aspergillosis in Severe COVID-19 and Possible Therapeutic Targets"

_diagnostics, 2022, doi:10.3390/diagnostics12061364_

Round 1
Reviewer 1 Report
This article reviews covid19, IPA and CPAP in detail from the aspects of cytokines and related targeted therapy, which plays a great guiding role in clinical diagnosis and treatment.
The main question addressed by the research is CAPA. It is very interesting. The review analysises cytokine storm in severe COVID-19 and cytokine expression in IPA, and describes genetic susceptibility to Aspergillus infection, shows that the same cytokines between severe COVID19 and IPA (i.e. high levels of TNF-α, IL-1, IL-6, IL-8, IL-10 and low levels of IFN-γ). And IL-6 and IL-8 are the assessment of CAPA risk , that play a role to stratify patients at higher and lower risk of fungal superinfection.It’s important that management of steroid therapy is a milestone of CAPA treatment suggesting that a lower dosage of steroids (dexamethasone 6 mg/daily) and an early de-escalation should be considered in patients with CAPA. Targeted therapies can be the future researchs. This article has a great guiding role in the clinical diagnosis and treatment of CAPA. The topic is original. The paper is well written.The text is clear and easy to read. The conclusions are consistent with the arguments presented. They address the main question posed. So I accept.
Author Response
Thank you very much for your support!
Reviewer 2 Report
This is a comprehensive narrative review on the relationship between the immune impairment related to covid-19 and the development of Invasive aspergillosis. The paper focus on the cytokine dysregulation in invasive Aspergillus infection and covid-19, and in preventive strategies.My only suggestion to the authors, given the variable prevalence and mortality of CAPA reported in the literature, is to describe in more details the diagnostic strategies for this condition, and the potential value of using combination of serum biomarkers and direct test such as culture and PCR (now included in the last EORTC/MSG criteria for IFI)
Author Response
My only suggestion to the authors, given the variable prevalence and mortality of CAPA reported in the literature, is to describe in more details the diagnostic strategies for this condition, and the potential value of using combination of serum biomarkers and direct test such as culture and PCR (now included in the last EORTC/MSG criteria for IFI)
R: dear reviewer, thank you very much for your suggestions. We included a paragraph on diagnostic strategies in conclusions section.
Reviewer 3 Report
It was a pleasure to review the paper “Cytokine Profile of Invasive Pulmonary Aspergillosis in Severe COVID-19 and Possible Therapeutic Targets”. The authors describe in an interesting and comprehensive way the issues related to COVID-19 associated pulmonary aspergillos. The literature used in the work is new and appropriate. The work also contains accurate summaries. The work deserves publication in Diagnostics.
The discussed topic concerning the cytokine profile from COVID-19 associated pulmonary aspergillos is very interesting and important. The information in this review may influence the treatment of this disease.
As we know, there are many publications on COVID-19 and its complications, as well as those on pulmonary aspergillos. The combination of these two topics and the compilation of relevant conclusions is innovative.
The work is written very well. Moreover, it is understandable and accessible to the reader.
The work contains a good summary and conclusions.
Author Response
Thank you very much for your support!